# Cricothyroid Dysfunction in Unilateral Vocal Fold Paralysis Females Impairs Lexical Tone Production

**DOI:** 10.3390/jcm11216442

**Published:** 2022-10-30

**Authors:** Yu-Cheng Wu, Tuan-Jen Fang, Hsiu-Feng Chuang, Alice M. K. Wong, Yu-Cheng Pei

**Affiliations:** 1Department of Physical Medicine and Rehabilitation, Chang Gung Memorial Hospital at Linkou, 5 Fushing St., Taoyuan 333, Taiwan; 2Department of Medical Education, Far Eastern Memorial Hospital, No. 21, Sec. 2, Nanya S. Rd., New Taipei City 220, Taiwan; 3Department of Otolaryngology, Chang Gung Memorial Hospital at Linkou, 5 Fushing St., Taoyuan 333, Taiwan; 4School of Medicine, Chang Gung University, 259 Wen-Hwa 1st Road, Taoyuan 333, Taiwan; 5Graduate School of Science Design Program in Innovation for Smart Medicine, Chang Gung University, 259 Wen-Hwa 1st Road, Taoyuan 333, Taiwan; 6Center of Vascularized Tissue Allograft, Chang Gung Memorial Hospital at Linkou, No. 5 Fushing St., Taoyuan 333, Taiwan

**Keywords:** unilateral vocal fold paralysis, cricothyroid muscle, Mandarin, tonal language, voice tone, fundamental frequency, laryngeal electromyography

## Abstract

In this cross-sectional study, we compared voice tone and activities relating to the laryngeal muscle between unilateral vocal fold paralysis (UVFP) patients with and without cricothyroid (CT) muscle dysfunction to define how CT dysfunction affects language tone. Eighty-eight female surgery-related UVFP patients were recruited and received acoustic voice analysis and laryngeal electromyography (LEMG) when the patient was producing the four Mandarin tones. The statistical analysis was compared between UVFP patients with (CT+ group, 17 patients) and without CT muscle (CT− group, 71 patients) involvement. When producing Mandarin Tone 2, the voice tone in the CT+ group had smaller rise range (*p* = 0.007), lower rise rate (*p* = 0.002), and lower fundamental frequency (F0) at the offset point of the voice (*p* = 0.023). When producing Mandarin Tone 4, the voice tone in the CT+ group had smaller drop range (*p* = 0.019), lower drop rate (*p* = 0.005), and lower F0 at voice onset (*p* = 0.025). The CT+ group had significantly lower CT muscle activity when producing the four Mandarin tones. In conclusion, CT dysfunction causes a limitation of high-rising tone in Tone 2 and high-falling tone in Tone 4, a property that dramatically limits the tonal characteristics in Mandarin, a tonal language. This limitation could further impair the patient’s communication ability.

## 1. Introduction

Unilateral vocal fold paralysis (UVFP), resulting from injuries of the recurrent laryngeal nerve (RLN), frequently occurs after thyroid, cervical spine, esophageal, and lung surgeries. The RLN innervates the thyroarytenoid–lateral cricoarytenoid (TA-LCA) muscle complex, a vocal fold adductor, during phonation. Accordingly, UVFP would thus induce dysphonia by limiting the function of vocal fold coordination, changing the glottal conformation during phonation and impeding the patient’s communication ability [1,2,3,4].

UVFP patients partly undergo both RLN and external branch of the superior laryngeal nerve (eSLN) injury. The eSLN controls the cricothyroid (CT) muscle long deemed a vocal fold tension controller. Activation of the CT muscles can increase tension, further producing a higher-pitched voice. However, the impact of CT muscle impairment on voice production in UVFP patients remains controversial. Several observational studies have reported that patients with dual neuropathy of RLN and eSLN tend to have a wider glottal gap, indicating that CT muscle impairment may affect the vocal fold position in UVFP patients [5,6]. Our previous study found that UVFP patients with coexisting CT muscle paralysis had a lower magnitude of vocal fold vibration, more jitter and shimmer in acoustic voice analysis, and worse voice-related quality of life [7,8]. These findings imply that the CT muscle plays a functional role in phonation, and UVFP patients with CT muscle impairment might have poorer voice tone control.

The speaking languages we use throughout the world can be divided into tonal and non-tonal ones according to the ability of the lexical tone to constrain lexical access [9,10,11]. That is, tone variations mainly reveal speakers’ emotional status in non-tonal languages [12], while lexical tone in tonal languages plays a vital role in constraining spoken word identity and is thus vital for spoken word identification.

As a tonal language, Mandarin is unique, as it phonetically distinguishes four tones [13,14], each of which has a distinctive fundamental frequency (F0) contour and can thus be distinguished by listening. Tone 1 has a monotonic pitch, Tone 2 has a high-rising pitch, Tone 3 has a prolonged phonation and descending pitch, and Tone 4 has a short phonation, with a sharply descending pitch. In Mandarin, the same segmental context carries different meanings depending on the tone, and the ability to produce Mandarin tones is fundamental for successful communication in daily living [15]. To this end, we hypothesize that impairment of voice tone control in UVFP patients with CT muscle dysfunction may further decrease their conversational intelligibility.

In the present study, we investigated the degree to which the production of lexical tones is impaired by coexisting CT muscle dysfunction in UVPF. To this end, the patient’s voice and muscle activities were simultaneously recorded when producing the voice. To exclude the influence of sex, we recruited female patients with UVFP iatrogenically caused by surgery, a patient group that is more homogenous in their disease nature [3,16] and the severity of denervation [17], with better test-retest reliability for the upwards glissando sound [18]. As voice tone control is critical in speaking Mandarin, we hypothesize that the CT muscle dysfunction in UVFP could further impair the produced lexical tones.

## 2. Materials and Methods

### 2.1. Human Subjects

Subjects were recruited from a referral voice center from March 2015 to December 2020. The inclusion criteria were female patients with symptomatic UVFP occurring immediately after surgery. The diagnosis was confirmed by unilateral vocal fold paralysis observed by videolaryngostroboscopy and denervation changes in the unilateral TA-LCA muscle complex observed by needle laryngeal EMG. The exclusion criteria were patients with a prior history of vocal fold paralysis, not cooperating with assessments, incapable of speaking Mandarin, and with normal thyroarytenoid muscles or abnormal signals on both vocal folds confirmed by laryngeal EMG.

### 2.2. Procedures

Patients underwent assessments, including functional laryngeal EMG and acoustic voice analysis. The interval between the date of the surgery and the date of laryngeal EMG was calculated.

### 2.3. Real-Time Mandarin Fundamental Frequency Assessment

Following the instructor, the patient was asked to repeat the voiced bilabial nasal in Mandarin for four tones in sequence three times. The voiced bilabial nasal was chosen for its relatively simple articulation mechanism.

An automatic algorithm using MATLAB (The MathWorks, Natick, MA, USA) was developed to assess the time spectrogram through Fourier transform, the temporal dynamics of fundamental frequency, and voice energy.

The voice energy was represented by the root-mean-square value of the recorded voice. Along with the time domain, an abrupt increase in voice energy was used to detect the beginning of the voice, and a return to baseline level indicated the ending of the voice. In the frequency domain, harmonic series of voices were derived based on their corresponding absolute values of the power vector. The fundamental frequency was defined as the lowest frequency with the highest voice energy among the accompanied frequencies. Finally, for each patient, the results of the three trials were averaged.

To measure the characteristics of the four tones, we defined a variety of temporal, frequency, and temporal-frequency parameters. Figure 1 presents a schematic diagram of the points of these parameters. Our algorithm detected these points in each Mandarin lexical tone. In Mandarin Tone 1, the tone with monotonic pitch, we analyzed the change in F0 from onset to offset ($\Delta F0_{\bar{ON\;OFF}})$ to measure the levelness of the tone. In Tone 2, the high-rising tone, we analyzed the change in F0 from onset to offset ($\Delta F0_{\bar{ON\;OFF}})$ to evaluate the rise in the tone. In Tone 3, the tone with decreasing and then rising pitch, the changes in F0 from onset to offset ($\Delta F0_{\bar{ON\;OFF}})$, from onset to minimal F0 point ($\Delta F0_{\bar{ON\;MIN}})$, and from minimal F0 point to offset ($\Delta F0_{\bar{MIN\;OFF}})$ were chosen as characteristics. In Tone 4, the high-falling tone, the change in F0 from onset to offset ($\Delta F0_{\bar{ON\;OFF}})$ and maximum F0 drop in 5 ms (5 ms is the moving distance of the Fourier transform sliding window in the time domain, which is the minimal time unit in our analysis) were deemed contour features for fall measurement. In addition to the aforementioned absolute value of F0 change, we also analyzed each voice segment duration and the slope of change by dividing the absolute value of F0 change by the duration of each voice segment.

### 2.4. Functional Laryngeal EMG

The automatic program we developed can also analyze raw electromyography (EMG) data to yield instantaneous recruitment for laryngeal muscles. The raw EMG waveforms were first binned into non-overlapping epochs. The epoch duration for the CT muscles was 50 ms [7]. Each turn’s timing and amplitude were localized using the automatic algorithm. Specifically, we defined a turn as the change in polarity with an amplitude of at least 100 µV before and after the change to exclude noise-related peaks. Turn frequency was computed for each epoch as the number of turns divided by the epoch duration. Peak turn frequency was each muscle’s highest epoch turn frequency while phonating each tone.

### 2.5. Statistical Analysis

The intraclass correlation coefficient and cross-trial standard deviation were used to measure the test-retest reliability of voice acoustic analysis parameters. Differences between the CT+ and CT− groups were compared using chi-squared tests for nominal data (such as lesion side and etiology) and Student’s t-tests for numerical data. Cohen’s *d* was applied to represent the effect size. In the statistical analysis for each parameter, we only used complete data. The level of significance was defined as *p* < 0.05.

## 3. Results

A total of 117 female patients with dysphonia were first recruited, of whom 29 were excluded because ten patients had laryngeal EMG results incompatible with unilateral vocal fold paralysis (four with typical TA-LCA muscle complex, one with bilateral TA-LCA muscle complex impairment, and five with bilateral CT muscle impairment) and 19 patients had incomplete data (laryngeal EMG or acoustic voice data) (Figure 2). Among the 88 included patients, 17 had CT involvement (CT+ group), and the remaining 71 did not (CT− group). Table 1 shows patient demographics and the etiology of UVFP. There were no significant differences in age (*p* = 0.659), time after paralysis (*p* = 0.217), etiology (*p* = 0.112), or the side of vocal fold paralysis (*p* = 0.099) between the CT+ and CT− groups.

The upper row of Figure 3 shows the acoustic results of a sample patient in the CT+ group (hereafter referred to as ‘Patient A’) and a sample patient in the CT− group (hereafter referred to as ‘Patient B’). The F0 of Patient A is lower than that of Patient B among different Mandarin tones. Notably, the F0 of Patient A shows a limited range of dynamics compared with Patient B, especially in Tone 4, a tone featuring a sharply decreased pitch.

We then examined whether the findings that CT+ group patients tend to have a limited dynamics of F0 could also be found in the mean across a population of patient subjects. The lower row of Figure 3 compares the F0 average of four Mandarin tones in CT+ and CT− group patients. This figure shows that the patients in the CT+ group had smaller rise and lower rise rate in Mandarin Tone 2 and smaller drop and lower drop rate in Mandarin Tone 4. In Tone 1, the fundamental frequency in the CT+ group appears to be lower than that in the CT− group. Table 2 compares the acoustic features between the CT+ and CT− groups, an approach that can objectively reveal the difference in F0 contour between the two groups. When producing Mandarin Tone 1, the CT+ group had a lower F0 at the onset point of the voice than the CT− group (CT+ vs. CT−: 212.3 ± 41.5 vs. 233.4 ± 39.3, *p* = 0.035, Cohen’s *d* = 0.53). When producing Mandarin Tone 2, the voice tone in the CT+ group had smaller rise (CT+ vs. CT−: 24.6 ± 23.8 vs. 42.9 ± 33.3, *p* = 0.007, Cohen’s *d* = 0.57), smaller rise per second (CT+ vs. CT−: 30.7 ± 27.3 vs. 56.6 ± 44.9, *p* = 0.002, Cohen’s *d* = 0.61), and lower F0 at the offset of the voice (CT+ vs. CT−: 218.7 ± 54.1 vs. 250.0 ± 54.9, *p* = 0.023, Cohen’s *d* = 0.56). In Mandarin Tone 3, there was no difference between the two groups in the absolute value of F0 change, its slope from onset to offset and its slope from onset to the minimal F0 point. The CT+ group had smaller rise (CT+ vs. CT−: 8.9 ± 16.3 vs. 18.2 ± 28.8, *p* = 0.041) and lower rise rate from the minimal F0 point to offset (CT+ vs. CT−: 20.4 ± 32.0 vs. 49.1 ± 66.5, *p* = 0.006), while their effect sizes were small ($\Delta F0_{\bar{MIN\;OFF}}$ Cohen’s *d*: 0.35, $\Delta F0_{\bar{MIN\;OFF}}/\Delta T$ Cohen’s *d*: 0.47). When producing Mandarin Tone 4, the voice tone in the CT+ group had smaller drop (CT+ vs. CT−: −53.2 ± 28.3 vs. −70.8 ± 36.3, *p* = 0.019, Cohen’s *d* = 0.50), lower drop rate (CT+ vs. CT−: −125.9 ± 68.0 vs. −179.1 ± 87.0, *p* = 0.005, Cohen’s *d* = 0.64), lower F0 at the onset of the voice (CT+ vs. CT−: 231.0 ± 51.3 vs. 259.2 ± 49.6, *p* = 0.025, Cohen’s *d* = 0.57), and a much lower maximum F0 drop in 5 ms (CT+ vs. CT−: 16.1 ± 5.7 vs. 27.8 ± 14.8, *p* < 0.001, Cohen’s *d* = 0.86).

Table 3 shows the test-retest reliability of the temporal and frequency parameters in the acoustic voice analysis. We calculated the cross-trial STD and intraclass correlation coefficient across the three trials. The results showed that the cross-trial STD was relatively small compared with the mean value for each parameter. Importantly, all intraclass correlation coefficients (ICCs) were > 0.9 except that the $\Delta F0_{\bar{ON\;OFF}}$ (ICC = 0.77) of Tone 4. These results indicate good-to-excellent test-retest reliability in all temporal and frequency parameters of the acoustic voice analysis.

In terms of time duration, for Tone 1, the CT+ and CT− groups had durations of 0.85 s and 0.81 s, respectively (*p* = 0.219). The CT+ and CT− groups did not differ in their voice duration for Tone 1, Tone 2 (0.87 s vs. 0.84 s, *p* = 0.276), Tone 3 (0.70 s vs. 0.72 s, *p* = 0.379), or Tone 4 (0.46 s vs. 0.45 s, *p* = 0.383), indicating that voice duration is not affected by CT muscle involvement.

The results of functional laryngeal electromyography of the CT+ and CT− groups are shown in Table 4. The peak turn frequency of the affected side CT muscle in the CT+ group is significantly lower than that in the CT− group when producing each of the four Mandarin tones. When phonating Mandarin Tone 1, the peak turn frequency of the CT+ and CT− groups was 442.4 Hz and 720.6 Hz (*p* < 0.001, Cohen’s *d* = 1.19). From Tones 2 to 4, the peak turn frequencies of the CT+ and CT− groups were 439.6 Hz and 666.7 Hz (*p* = 0.001, Cohen’s *d* = 1.02), 381.6 Hz and 594.4 Hz (*p* = 0.003, Cohen’s *d* = 0.92), and 457.6 Hz and 803.0 Hz (*p* < 0.001, Cohen’s *d* = 1.41), respectively.

## 4. Discussion

This study investigated the role of CT muscle involvement in voice tone in female patients with UVFP. The CT muscle is generally regarded as a vocal fold tension controller that increases the tension of vocal folds during phonation. Previous studies have shown a strong predominance of CT muscle activation in the pitch-raising mechanism [19,20]. This phenomenon was found not only when speaking English [21] but also in phonating Japanese [22,23], Thai [24], Swedish [25], Dutch [26], and Danish [27], implying that the role of the CT muscle in F0 raising is cross-lingual. Therefore, dysfunction of the CT muscle is generally deemed to result in decreased fundamental frequency regardless of the language spoken [28,29]. The altered fundamental frequency caused by CT muscle dysfunction may further impair communication ability, especially in Mandarin, because lexical tone has been reported to be able to influence spoken word recognition in Mandarin significantly [30]. Zou et al. [31] also found that tone violation can dramatically increase listening comprehension error rates in Mandarin, even more than rhyme violation.

We successfully developed an autonomic F0 analysis algorithm for the four Mandarin tones with excellent test-retest reliability with respect to the measured parameters. The results indicate that, when producing Mandarin Tone 2, the high-rising tone, the CT+ group had a smaller rise, a lower rise rate, and a lower F0 at the offset point of voice. These findings support the hypothesis that CT muscle impairment causes a limited increase in frequency in the CT+ group. Likewise, in Mandarin Tone 4, the high-falling tone, the CT+ group had a smaller drop, a lower drop rate, and a lower F0 at the onset point of the voice, supporting the hypothesis that CT muscle impairment causes difficulty in phonating high pitch by limiting the onset F0 in Tone 4. These findings are compatible with Karen Ann Kochis-Jennings’s finding that CT muscle activities were comparable to TA/LCA activities when producing tones >300 Hz, while CT muscle activities were much lower than TA/LCA activities when producing tones <300 Hz [32], again indicating the impact of CT muscle impairment on high-pitched sound phonations.

The findings in Mandarin Tones 2 and 4 indicate that CT muscle impairment limits the phonation of high-pitched sounds, further narrowing the F0 ranges. McGarr and Osberger [33] first suggested that impairment of intonation may cause poor intelligibility, and Kent and Rosenbek [34] further hypothesized that reduced F0 variation is the cause of poor intelligibility. This hypothesis was supported by studies using resynthesized speech that applied a flattened F0 contour of speech obtained from typical speakers [35,36,37,38] and those with dysarthria [39]. These findings suggest that the decrease in F0 ranges in Tones 2 and 4 impairs conversational intelligibility in the CT+ patients.

Laures and Weismer [36] provided three possible explanations for poor speech understanding with a flattened intonation contour. First, intonation directs listeners to important words, expending more processing priority. Diminishing intonation presents a greater difficulty for listeners when comprehending high-content components of an utterance, since the cues of their locations are deleted. Second, decrement of dynamic change in F0 interferes with the segmentation of words in continuous speech, which complicates the task of parsing speech into meaning units. In consequence, intelligibility suffers. The last explanation is based on the sufficient contrast hypothesis [40]. This hypothesis presupposes that vowel intelligibility decreases with higher F0 due to the relatively wider spacing within source harmonics, causing an under-sampled spectral envelope of formant peaks. Likewise, a flat F0 may reduce the density of harmonics within a formant peak, affecting intelligibility due to the under-sampled spectral envelope, which can reduce formant peaks in the amplitude spectrum [41] and further decrease the local signal-to-noise ratio in noisy backgrounds [37]. The signal-to-noise ratio is a primary determinant of detection and discrimination thresholds in noise [42]. A decrease in the signal-to-noise ratio may then reduce performance in understanding speech.

The impaired speaking intelligibility caused by flattened the intonation contour in English, a non-tonal language, may also occur in tonal languages, since lexical tone in tonal languages serves the same function as that in non-tonal languages when it comes to revealing emotional status and segmentation. Besides these functions, lexical tone in tonal languages also plays a vital role in constraining spoken word identity, making tone identification much more critical in the comprehension of tonal languages [15]. Therefore, flattening intonation contours may make lexical tone less distinguishable. For example, Mandarin Tones 2 and 4 in the CT+ group in our study were not as high-rising or high-falling as those in the CT− group. This phenomenon may give listeners difficulties when telling Mandarin Tone 2 from Mandarin Tone 4. Given that lexical tone is essential for constraining semantic meaning in Mandarin, the limitation of producing an accurate lexical tone will further hinder the patient’s communication ability.

In addition, in functional laryngeal EMG analysis, the peak turn frequency of the affected CT muscle in the CT+ group was significantly lower than the ipsilateral one in the CT− group when phonating the four Mandarin tones. A previous study from our team found that quantitative laryngeal EMG of the CT muscle when making an upwards glissando sound can reflect the level of SLN injury in patients with UVFP [18]. The present study found a similar phenomenon when phonating the four Mandarin tones, implying a broader possibility of using peak turn frequency to predict CT muscle function in UVFP patients.

There were several limitations to this study. First, because gender differences themselves can influence the fundamental frequencies, we only recruited female surgery-related UVFP patients. In the future, we will also analyze acoustic voice data in other patient groups, such as male or nonsurgical-related UVFP patients, to examine whether the impact of CT muscle impairment could differ between sexes or among etiologies. Second, we could not conduct a subgroup analysis to compare acoustic voice data between CT muscle impairment etiologies because of a limited case number. UVFP patients with a different cause of CT muscle impairment may have different voice performance characteristics. More patients should be enrolled in future works to clarify this issue.

## 5. Conclusions

To the best of our knowledge, this study is the first to reveal the impact of CT dysfunction on Mandarin tone phonation in UVFP patients. Previous literature revealed a strong cross-lingual predominance of CT muscle activation in the pitch-raising mechanism and the influence of lexical tone on spoken word recognition in Mandarin. We successfully developed an autonomic F0 analysis algorithm for the four Mandarin tones with excellent test-retest reliability with respect to the measured parameters. We found that, in female surgery-related UVFP patients, CT muscle impairment can limit the rise in Mandarin Tone 2 and the fall in Tone 4 by separately lowering the offset and onset point F0 when phonating Tone 2 and Tone 4, making the lexical tone of Mandarin in these patients less distinguishable and further impeding their communication function. In the functional laryngeal EMG analysis, the peak turn frequency of impaired CT muscle was found to be lower than that of normal muscle, implying a possibility of using peak turn frequency to predict CT muscle function in UVFP patients.

## Figures and Tables

**Figure 1 jcm-11-06442-f001:**
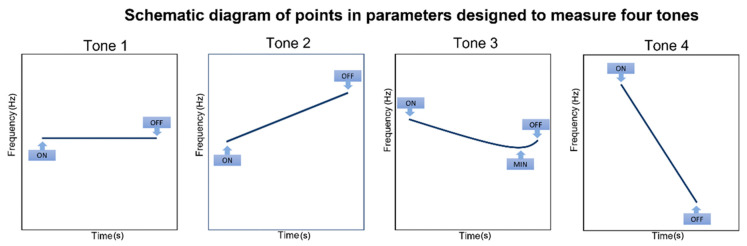
Schematic diagram of points in parameters designed to measure four tones. Each panel shows F0 as a function of normalized time. The blue lines from left to right refer to F0 from Tone 1 to Tone 4. Point ON is the onset point of voice, point OFF is the offset point, and point MIN has minimal tone frequency.

**Figure 2 jcm-11-06442-f002:**
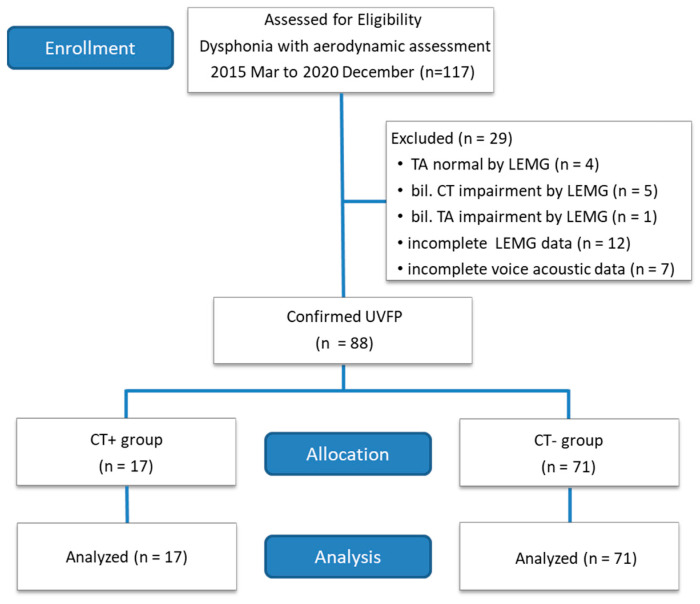
Flowchart of patient recruitment, exclusion, group assignment, and analysis. TA = thyroarytenoid–lateral cricoarytenoid muscle complex; CT = cricothyroid muscle; LEMG = laryngeal electromyography.

**Figure 3 jcm-11-06442-f003:**
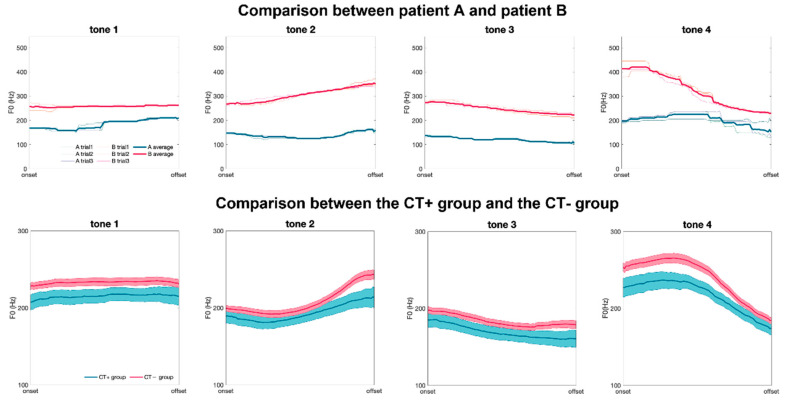
Comparison of acoustic voice analysis results. Each panel shows F0 as a function of normalized time, from the beginning to the end of each word. Panels from left to right represent the F0 of tones 1 to 4 in sequence. The upper row is the comparison between Patients A and B. The three thin blue and red lines represent the F0 contours of the three trials of Patients A and B, respectively, and the bold lines represent the means of the three trials for each patient. The lower row compares the CT+ group (blue band) and CT− group (red band). The bold line is the average of F0, and the width of each band represents the standard error of the mean (SEM).

**Table 1 jcm-11-06442-t001:** Patient characteristics of the study groups.

Parameter	Total	Group		
		CT+	CT−	*p* Value
	n = 88	n = 17	n = 71	
Age (year)	52.15 ± 14.05	50.76 ± 13.92	52.50 ± 14.06	0.659
Paralysis side (left/right)	49/33	7/10	47/24	0.099
Time post-paralysis (month)	9.56 ± 36.05	4.52 ± 3.10	10.88 ± 40.36	0.217
Pathogenesis (n, %)				0.112
Thyroidectomy	51 (62.2)	13 (76.5)	43 (60.6)	
Esophageal surgery	5 (6.1)	0 (0)	5 (7.0)	
Lung surgery	10 (12.2)	0 (0)	10(14.1)	
Skull base or brain surgery	3 (3.7)	2 (11.8)	1 (1.4)	
Cervical spine surgery	7 (8.5)	2 (11.8)	5 (7.0)	
Heart surgery	4 (4.9)	0 (0)	4 (5.6)	
Other	2 (2.4)	0 (0)	3 (4.2)	

Data are presented as the mean ± standard deviation or number (percentage).

**Table 2 jcm-11-06442-t002:** Comparison of fundamental frequency (F0) between the CT+ and CT− groups when producing the four tones.

F0(Hz)	Total	Group		*p* Value	Cohen’s *d*
		CT+	CT−		
Tone 1					
Onset	229.3 ± 40.4	212.3 ± 41.5	233.4 ± 39.3	0.035 *	0.53
Offset	233.3 ± 46.9	219.8 ± 48.6	236.5 ± 46.2	0.106	0.35
$\;\;\Delta F0_{\bar{ON\;OFF}}$	3.9 ± 17.1	7.5 ± 22.0	3.1 ± 15.8	0.219	0.26
$\;\;\Delta F0_{\bar{ON\;OFF}}/\Delta T$	5.1 ± 23.8	8.3 ± 28.7	4.3 ± 22.6	0.298	0.17
Tone 2					
Onset	204.3 ± 37.5	194.1 ± 38.6	206.7 ± 37.1	0.118	0.33
Offset	243.6 ± 55.8	218.7 ± 54.1	250.0 ± 54.9	0.023 *	0.56
$\;\;\Delta F0_{\bar{ON\;OFF}}$	39.3 ± 32.4	24.6 ± 23.8	42.9 ± 33.3	0.007 **	0.57
$\;\;\Delta F0_{\bar{ON\;OFF}}/\Delta T$	51.6 ± 43.2	30.7 ± 27.3	56.6 ± 44.9	0.002 **	0.61
Tone 3					
Onset	202.1 ± 38.6	190.5 ± 39.5	204.9 ± 38.0	0.093	0.37
Offset	184.2 ± 49.9	165.7 ± 46.2	188.7 ± 50.0	0.041 *	0.46
$\;\;\Delta F0_{\bar{ON\;OFF}}$	−17.9 ± 40.0	−24.8 ± 31.5	−16.2 ± 32.1	0.162	0.27
$\;\;\Delta F0_{\bar{ON\;OFF}}/\Delta T$	−36.5 ± 54.0	−48.9 ± 70.0	−33.5 ± 50.0	0.200	0.29
$\;\;\Delta F0_{\bar{ON\;MIN}}$	−34.3 ± 18.2	−33.7 ± 21.6	−34.5 ± 17.4	0.449	0.03
$\;\;\Delta F0_{\bar{ON\;MIN}}/\Delta T$	−100.3 ± 72.7	−82.5 ± 62.2	−105.9 ± 74.2	0.095	0.33
$\;\;\Delta F0_{\bar{MIN\;OFF}}$	16.4 ± 27.0	8.9 ± 16.3	18.2 ± 28.8	0.041 *	0.35
$\;\;\Delta F0_{\bar{MIN\;OFF}}/\Delta T$	43.6 ± 62.2	20.4 ± 32.0	49.1 ± 66.5	0.006 **	0.47
Tone 4					
Onset	253.8 ± 50.9	231.0 ± 51.3	259.2 ± 49.6	0.025 *	0.57
Offset	186.4 ± 37.2	177.8 ± 34.0	188.5 ± 37.9	0.131	0.29
$\;\;\Delta F0_{\bar{ON\;OFF}}$	−67.4 ± 35.4	−53.2 ± 28.3	−70.8 ± 36.3	0.019 *	0.50
$\;\;\Delta F0_{\bar{ON\;OFF}}/\Delta T$	−168.9 ± 85.9	−125.9 ± 68.0	−179.1 ± 87.0	0.005 **	0.64
Maximal drop in 0.005 s	25.5 ± 14.3	16.1 ± 5.7	27.8 ± 14.8	<0.001 ***	0.86

* *p* < 0.05; ** *p* < 0.01; *** *p* < 0.001. F0, fundamental frequency; T, time; ON, onset point of voice; OFF, offset point of voice; MIN, minimal fundamental frequency point. $\Delta F0_{\bar{ON\;OFF}}\;$= change in F0 from onset to offset; $\Delta F0_{\bar{ON\;OFF}}/\Delta T$= change in F0 from onset to offset divided by duration from onset to offset.

**Table 3 jcm-11-06442-t003:** Test-retest reliability of temporal and frequency parameters in the acoustic voice analysis.

Parameters (Hz)	Mean ± STD	Cross-Trial STD	Intraclass Correlation Coefficient
Tone 1			
Onset	229.3 ± 40.4	9.5	0.99 **
Offset	233.3 ± 46.9	10.9	0.98 **
$\;\;\Delta F0_{\bar{ON\;OFF}}$	3.9 ± 17.1	11.3	0.90 **
Time duration	0.82 ± 0.18	0.08	0.95 **
Tone 2			
Onset	204.3 ± 37.5	8.7	0.99 **
Offset	243.6 ± 55.8	14.0	0.99 **
$\;\;\Delta F0_{\bar{ON\;OFF}}$	39.3 ± 32.4	15.1	0.94 **
Time duration	0.84 ± 0.17	0.08	0.90 **
Tone 3			
Onset	202.1 ± 38.6	9.5	0.97 **
Offset	184.2 ± 49.9	13.3	0.99 **
$\;\;\Delta F0_{\bar{ON\;OFF}}$	−17.9 ± 40.0	16.5	0.93 **
Time duration	0.71 ± 0.17	0.09	0.95 **
Tone 4			
Onset	253.8 ± 50.9	12.2	0.97 **
Offset	186.4 ± 37.2	14.0	0.95 **
$\;\;\Delta F0_{\bar{ON\;OFF}}$	−67.4 ± 35.4	15.7	0.77 *
Time duration	0.45 ± 0.11	0.05	0.92 **

STD, standard deviation; F0, fundamental frequency; T, time; ON, onset point of voice; OFF, offset point of voice; $\Delta F0_{\bar{ON\;OFF}}$ = change of F0 from onset to offset; ICC = intraclass correlation coefficient * ICC > 0.75; ** ICC > 0.9.

**Table 4 jcm-11-06442-t004:** Comparison of peak turn frequency between the CT+ and CT− groups when producing the four tones.

Peak Turn Frequency in Lesion Site (Hz)	Total	Group		*p* Value	Cohen’s *d*
		CT+	CT−		
Tone 1	674.9 ± 266.8	442.4 ± 285.0	730.6 ± 231.6	<0.001 ***	1.19
Tone 2	622.9 ± 238.4	439.6 ± 252.5	666.7 ± 214.3	0.001 **	1.02
Tone 3	553.3 ± 245.0	381.6 ± 263.1	594.4 ± 223.5	0.003 **	0.92
Tone 4	736.3 ± 279.0	457.6 ± 287.5	803.0 ± 233.3	<0.001 ***	1.41

** *p* < 0.01; *** *p* < 0.001.

## Data Availability

The data presented in this study are available on request from the corresponding author.

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
