# Peer review of "Cricothyroid Dysfunction in Unilateral Vocal Fold Paralysis Females Impairs Lexical Tone Production"

_jcm, 2022, doi:10.3390/jcm11216442_

Round 1

Reviewer 1 Report

Manuscript Number: jcm-1988446-peer-review-v1

Title:  

 Cricothyroid dysfunction in unilateral vocal fold paralysis fe-2 males impairs lexical tone production

1. Yes, this subject is useful for publication in JCM
2. Authors compared voice tone and activities relating to the laryngeal muscle between unilateral vocal fold paralysis patients with and without cricothyroid muscle dysfunction to define how CT dysfunction affects language tone.

3. The design, methods, and results, statistics are clearly presented. Fig. 1 is not clear – what frequency Hz? and time sec?

4. Discussion is logical and correct.

5. Conclusion is too short.
6. References are current and pertinent.

This paper should be published after minor corrections.

Reviewer 2 Report

This paper is a prospective cross-sectional study that seeks to identify the contribution of cricothyroid-CT (superior laryngeal nerve) dysfunction in the presence of iatrogenic (surgery related) unilateral vocal paralysis-UVFP (recurrent laryngeal nerve). 88 female subjects were recruited of which 17 were also found to have involvement of the CT muscle. Acoustic voice measures and laryngeal electromyography-EMG were assessed with the 4 Mandarin vocal tones. The authors found tones 2 (high-rising) and 4 (high-falling) were significantly affected as measured by these parameters.

The paper is well-written and experimental methods nicely described. It is also thoughtful that they include the time post paralysis, which would have potential relevance to the findings. The statistical analysis in Table 2 adds to the value of the paper. This study adds to the existing literature by quantifying the functional defects of the CT muscle in Mandarin tones, which has previously been performed in multiple other languages. The findings of CT effect on speech is expected based on the previous literature.

This is a nice scientific study, adding clinical relevance to the Mandarin-speaking population. The authors do not elaborate on clinical, prognostic or interventional significance, which would strengthen its contribution to the clinical literature.

In Methods, consider moving the paragraph on voice centre recruitment lines 127-137 up to line 91, and delete the redundant statement on the Human Subject's Committee and informed consent.
